# Electrochemotherapy in Mucosal Cancer of the Head and Neck: A Systematic Review

**DOI:** 10.3390/cancers13061254

**Published:** 2021-03-12

**Authors:** Primož Strojan, Aleš Grošelj, Gregor Serša, Christina Caroline Plaschke, Jan B. Vermorken, Sandra Nuyts, Remco de Bree, Avraham Eisbruch, William M. Mendenhall, Robert Smee, Alfio Ferlito

**Affiliations:** 1Department of Radiation Oncology, Institute of Oncology Ljubljana and Faculty of Medicine, University of Ljubljana, 1000 Ljubljana, Slovenia; 2Department of Otorhinolaryngology and Cervicofacial Surgery, University Medical Centre Ljubljana and Faculty of Medicine, University of Ljubljana, 1000 Ljubljana, Slovenia; ales.groselj@kclj.si; 3Department of Experimental Oncology, Institute of Oncology Ljubljana and Faculty of Health Sciences, University of Primorska, Izola and Faculty of Health Sciences, University of Ljubljana, 1000 Ljubljana, Slovenia; gsersa@onko-i.si; 4Department of Otorhinolaryngology, Head & Neck Surgery and Audiology, Copenhagen University Hospital Rigshospitalet, 2100 Copenhagen, Denmark; caroline@dadlnet.dk; 5Faculty of Medicine and Health Sciences, University of Antwerp, Antwerp, and Department of Medical Oncology, Antwerp University Hospital, 2650 Edegem, Belgium; JanB.Vermorken@uza.be; 6Department of Oncology, KU Leuven, University of Leuven and Department of Radiation Oncology, UZ Leuven, 3000 Leuven, Belgium; sandra.nuyts@uzleuven.be; 7Department of Head and Neck Surgical Oncology, UMC Utrecht Cancer Center, University Medical Center Utrecht, Utrecht University, 3584 Utrecht, The Netherlands; R.deBree@umcutrecht.nl; 8Department of Radiation Oncology, University of Michigan, Ann Arbor, MI 48109-5010, USA; eisbruch@med.umich.edu; 9Department of Radiation Oncology, University of Florida, Gainesville, FL 32610-0385, USA; mendwm@shands.ufl.edu; 10Department of Radiation Oncology, The Prince of Wales Cancer Centre, Sydney, NSW 2031, Australia; Robert.Smee1@health.nsw.gov.au; 11Coordinator of the International Head and Neck Scientific Group, 35100 Padua, Italy; profalfioferlito@gmail.com

**Keywords:** electrochemotherapy, head and neck cancer, quality of life, systematic review

## Abstract

**Simple Summary:**

Electrochemotherapy is a topical ablative treatment based on the formation of electropores in the cell membrane exposed to an external electric field. The consequent intracellular accumulation of hydrophilic bleomycin or cisplatin molecules greatly increases their cytotoxicity. Currently, electrochemotherapy is recognized as an effective treatment for tumors of different histological types and also for some deep-seated tumors. In mucosal cancer of the head and neck, experience with electrochemotherapy is limited, primarily due to the anatomical complexity of the region and the poor accessibility of tumors, as well as the design limitations of the electrodes used to create an electric field. A systematic review of the literature and subsequent analysis of 164 patients from 16 studies treated between 1998 and 2020 confirmed that electrochemotherapy is an effective and safe treatment for mucosal cancer of the head and neck as well.

**Abstract:**

Electrochemotherapy (ECT) is a local ablative treatment that is based on the reversible electroporation and intracellular accumulation of hydrophilic drug molecules, which greatly increases their cytotoxicity. In mucosal head and neck cancer (HNC), experience with ECT is limited due to the poor accessibility of tumors. In order to review the experience with ECT in mucosal HNC, we undertook a systematic review of the literature. In 22 articles, published between 1998 and 2020, 16 studies with 164 patients were described. Curative and palliative intent treatment were given to 36 (22%) and 128 patients (78%), respectively. The majority of tumors were squamous cell carcinomas (79.3%) and located in the oral cavity (62.8%). In the curative intent group, complete response after one ECT treatment was achieved in 80.5% of the patients, and in the palliative intent group, the objective (complete and partial) response rate was 73.1% (31.2% and 41.9%). No serious adverse events were reported during or soon after ECT and late effects were rare (19 events in 17 patients). The quality-of-life assessments did not show a significant deterioration at 12 months post-ECT. Provided these preliminary data are confirmed in randomized controlled trials, ECT may be an interesting treatment option in selected patients with HNC not amenable to standard local treatment.

## 1. Introduction

Head and neck cancer (HNC) is the eighth most common and lethal cancer worldwide [1]. Despite the refinement of surgical and radiotherapy (RT) techniques and the introduction of new systemic drugs in recent decades, the results of existing treatments are not satisfying [2]. Disease-related and treatment-induced functional impairments (speech and swallowing), cosmetic disfigurement, and recurrent, metastatic, and second primary tumors arising in previously treated regions pose a challenge to clinicians dealing with this disease. In addition, due to the aging of the population, with a growing number of fragile elderly patients and an increasingly emphasized quality-of-life criterion, innovative treatment strategies are required [3].

Electrochemotherapy (ECT) is a topical ablative treatment based on the reversible electroporation, i.e., formation of permeable structures (i.e., electropores), in the cell membranes of tissue exposed to an external electric field [4]. Its execution is simple and described in the standard operating procedures: the patients are sedated or anesthetized and given either bleomycin or cisplatin intratumorally or intravenously [5,6]. After a short period, allowing the drug to distribute in the tumor, electric pulses are applied, using different electrode designs that cover the whole tumor mass, including the safety margins. Electroporation increases the cytotoxicity of bleomycin, and to a lesser extent cisplatin, dramatically. Currently, intravenously administered bleomycin is the most commonly used drug in ECT [6,7]. However, electroporation pulses also affect adjacent non-malignant cells in the tumor through other, indirect mechanisms, both vascular and immunological, which also contribute to the effectiveness of ECT [6]. The application of electric pulses to the tumor induces “vascular lock” by causing transient vasoconstriction and a decrease in blood flow in the treated tumor as a result of entrapment with a prolonged presence of drug molecules in the tumor. In addition, ECT affects the apoptosis of endothelial cells, leading to a vascular-disrupting effect [8]. The latter is selective toward tumor vessels, predominantly small ones, and does not affect normal vessels in the surrounding healthy tissue [9]. The immune effects are triggered by the immunogenic death of tumor cells. This effect contributes to the eradication of the remaining viable tumor cells, by eliciting local immune response. Therefore, ECT is considered an in situ vaccination, as some other ablative techniques, and can be effectively combined with immunotherapeutic approaches [10].

Especially after 2006, when the comprehensive European Standard Operating Procedures in Electrochemotherapy (ESOPE) guidelines were published, which made a decisive contribution to the standardization of ECT procedures in clinics, results have been accumulated on the efficacy of ECT in clinical oncology [5]. Currently, ECT is recognized as an effective topical treatment for tumors of different histological types [11,12]. It is primarily used in the treatment of skin cancers: basal cell and squamous cell carcinomas, cutaneous metastases of melanoma, Merkel cell carcinoma, and others [11,12,13]. However, ECT has also been studied in the treatment of deep-seated tumors, e.g., bone metastases, liver and pancreatic malignancies, prostate cancers, and gastrointestinal tumors [14]. In mucosal cancers of the head and neck, experience with ECT is limited, primarily due to the anatomical complexity of the region and the poor accessibility of tumors, as well as the design limitations of the electrodes used to create an electric field [14].

The purpose of this systematic review of the literature is to summarize the results of the treatment of mucosal HNC with ECT and to critically analyze its advantages, limitations, and future perspectives in the treatment of these tumors.

## 2. Materials and Methods

### 2.1. Search Strategy and Study Selection

A systematic search of the PubMed/Medline, Web of Sciences, Scopus, and Embase databases was conducted to collect published articles on ECT in mucosal HNC. In this review, the term “mucosal HNC” refers to tumors, primary or recurrent, located in the mucosal surface of the oral cavity, pharynx, larynx, nasal cavity, or paranasal sinuses. Malignancies arising from large salivary glands of the head and neck were excluded because they usually manifest as skin-infiltrating lesions. The following search terms were used: “electrochemotherapy” with synonyms (“electroporation,” “electroporation therapy,” and “electropermeabilization”) and “head and neck cancer” with related terms (“head and neck neoplasms” and “head and neck tumors”). The query was created by assigning the title, abstract, and keywords/MeSH fields to all terms and combining them with the Boolean operators OR and AND. Eligible manuscripts included full-length journal articles, case reports, letters to the editor, or short communications, written in English and published between 1 January 1991 and 30 October 2020. Unpublished studies, meeting abstracts, book chapters, and editorials were not included in this review.

Authors P.S. and A.G. independently reviewed the titles and abstracts to select studies for detailed reading and extraction of relevant data on patients, disease, and treatment characteristics and outcomes. Disagreements between the authors were resolved by discussion or arbitration with a third author (G.S.). Special attention was paid to the duplicated data appearing in several reports: in such cases, the more comprehensive study report with updated outcome information was used. In selected publications, the reference lists were screened to identify additional potentially eligible studies that were missed in the literature search.

Studies were included in this systematic review if they provided: (i) information about single-session ECT (anesthesia type, chemotherapeutic drug and route of its administration, and details on electric pulses and electrodes) of mucosal HNC using bleomycin or cisplatin administered intratumorally or intravenously, with curative or palliative intent; (ii) individual or group data on patients (gender and age), treated tumors (localization, primary/residual/recurrent, and stage and/or size), and tumor response. Information on toxicity and quality of life were also collected where available.

### 2.2. Statistics

The outcome measure of interest was the response of the individual lesion to ECT. For this purpose, the estimates given in the original articles were considered, regardless of the assessment method (clinical, histopathological, or radiological), evaluation criteria (Response Evaluation Criteria in Solid Tumors (RECIST) or World Health Organization (WHO)), or the time interval from the ECT to the response evaluation.

Basic descriptive statistics were reported with the medians, ranges, and interquartile ranges for numerical variables and as percentages for categorical variables. The association between two categorical variables was tested by the Fisher exact test, and two-sided p-values are reported.

## 3. Results

### 3.1. Search Results

The database search identified 249 records. After the removal of duplicates and non-target formats of publications, 155 titles were left for an abstract or a full text reading. Finally, 22 publications describing 16 studies with 164 patients were found to be appropriate for systematic review [15,16,17,18,19,20,21,22,23,24,25,26,27,28,29,30,31,32,33,34,35,36]. The selection procedure is shown in Figure 1.

### 3.2. Characteristics of the Eligible Studies

Most publications reported on studies of prospective design, i.e., phase 2 observational studies [17,18,19,20,22,24,25,26,27,29,30,31,33,34,35]; only three studies were retrospective [25,26,27], and one was a case report [36] (Table 1). Four studies were reported in more than one publication [15,16,17,21,23,25,28,29,32,33]. Individual patient data were reported in 10 studies (86 patients, 52.4%) [17,18,20,24,25,26,29,33,34,36] and group data in six studies (78 patients, 47.6%) [19,22,27,30,31,35]. The number of patients with mucosal HNC treated in these studies ranged from 1 to 43 patients (median 6.5; interquartile range 2.5–15.5). In five studies, only mucosal tumors were treated [18,29,31,34,36], whereas a combination of mucosal and cutaneous treatments was reported in 11 studies [17,19,20,22,24,25,26,27,30,33,35].

ECT was used in a curative setting in three studies, either in the preoperative phase as a neoadjuvant treatment in order to reduce the size of the tumor before definitive therapy, or as a first-line treatment for early-stage tumors [18,29,30]. The inclusion criteria in 12 studies were limited to progressive/recurrent and inoperable HNC in patients heavily pre-treated by multimodal therapy and/or who had been refused all available curative treatment options; in these studies, ECT was a palliative treatment [17,19,20,22,24,26,27,31,33,34,35,36]. In one study, curative- and palliative-intent ECT was employed [25].

### 3.3. Patients and Treated Tumors

Cohorts were composed predominantly of male patients (71.7%) and the median age of those with known age and in cohorts with age reported for the group ranged from 57 to 68 years (minimal and maximal age 20 and 87 years, respectively) (Table 2). Thirty-six patients (22%) were treated with curative intent and 128 patients (78%) for palliation.

The majority of tumors were squamous cell carcinomas (79.3%), located in the oral cavity (62.8%) or oropharynx (12.2%), although individual tumors were also treated at less accessible sites (e.g., three nasal cavity/paranasal sinuses, one hypopharynx, and two nasopharynx). Their extent was described either in terms of T-stage or size (cm), and included all stages from T1 to T4, with the largest diameters ranging from 1 to 14.5 cm.

### 3.4. Electrochemotherapy Treatment

Details on the treatment techniques are shown in Table 3. The vast majority of patients (105, 88.2%) received one ECT application, and only 10 and 4 patients underwent a second and a third course of ECT, respectively; the number of ECT courses applied was not specified in 45 cases. In 93.1% of the patients for whom pertinent information was available, ECT was performed under general anesthesia and bleomycin was used in all procedures. The predominant route of drug administration was intravenous (62.5%), employing a uniform dose of 15,000 international units (IU)/m^2^ [22,24,25,26,27,30,31,33,34,35,36]. Intratumoral injections were administered at a fixed dose of 1000 IU/cm3 [17,18,19,20,29] or at a range of between 250 and 1000 IU/cm^3^, depending on tumor size [22,27,31]. For the majority of applications (72.3%), hexagonal array electrodes were selected. Information on the inclusion of safety margins around the tumor lesion in the electric field was available for only two thirds of the cases (108, 65.9%). All margins were treated in 78 cases, and in 30 cases, only debulking of the accessible tumor part was performed due to excessive tumor size and/or inaccessibility to be covered with a sufficient electric field. The electric pulses were delivered by a Cliniporator (IGEA, Carpi, Italy) and the procedure was performed according to the ESOPE guidelines in all patients treated after 2006 (116, 70.7%).

### 3.5. Response to Electrochemotherapy and Outcome

#### 3.5.1. Curative Setting

All 36 patients had one cycle of ECT, 12 of them in combination with subsequent RT (57.8 Gy, 1.7 Gy twice a day, in 2.5 weeks) [29]. Complete response (CR) was recorded in 29 cases (80.5%), partial response (PR) in six cases (16.7%), and stable disease (SD) in one case (2.8%) [18,25,29,30]. All CRs were determined by histopathological examination of surgical specimens (10 tumors, four weeks after ECT) or biopsy samples (19 tumors, eight weeks after ECT (N = 7) or ECT + RT (N = 12)). All tumors except one were cT1–2 oral cavity or oropharyngeal squamous cell carcinomas; only the oral tongue cT2 primary was histologically adenocarcinoma [18,29]. Among the PR and SD cases, two were evaluated histopathologically (four weeks after ECT) and five clinically (six or eight weeks after ECT, using WHO or RECIST criteria) [18,25,30]. Histologically, all tumors were squamous cell carcinomas of stage cT2 (three cases) or cT3–4 (four cases); the latter had ECT for neoadjuvant debulking before chemoradiation [30]. Survival outcome was reported for 31 patients: no recurrence was observed locally in 29 complete responders to ECT and in two patients with partial response (evaluated histopathologically in surgical specimens) after follow-up ranging from 2 to 67 months (median 24, interquartile range 10.5–60.5) [18,29].

#### 3.5.2. Palliative Setting

The response to ECT was evaluated clinically, histologically, and/or radiologically at four to eight weeks after the procedure. It was specified in 106 of 128 patients: 32 were CRs (30.2%), 44 PRs (41.5%), 24 SDs (22.6%), and 6 progressive diseases (PDs) (5.7%) (Table 4). Statistically significantly more CRs were recorded in patients with T1–2 tumors compared to T3–4 tumors (52.2 vs. 12.5%, *p* = 0.005). Eight of 15 patients with CR and available information on survival were followed up after 12 months or more: two of them experienced local recurrence (one had salvage surgery and died of a stroke 32 months post-ECT) and the others were free of local failure from 12 to 42 months (median 27.5, interquartile range 14.5–37.5) [17,26,34]. Among 29 patients with non-CR after ECT, only nine patients survived 12 months or more. All but two patients died between 12 and 20 months (median 14, interquartile range 13.5–19.5) after treatment and two were alive with the disease at 12 and 14 months [17,25,34].

### 3.6. Electrochemotherapy-Related Toxicity and Quality of Life

Systematic evaluation of the different possible acute and late toxicities of ECT in terms of the National Cancer Institute Common Terminology Criteria for Adverse Events or comparable scale was used in only three studies [27,31,34], and in one study serious treatment-related adverse events were pre-specified [29]. Separately and more often, pain was assessed by using WHO criteria [18], analgesia post-surgery cards (which include the visual analog scale (VAS) and the verbal rating scale (VRS)) [30], the numeric rating scale (NRS) [31,34], the VAS [33,35,36], or other quality-of-life questionnaires [29,30,31,34]. Similarly, bleeding was systematically evaluated in one study [35]. Especially in older studies, only a descriptive assessment of toxicity was recorded [17,18,19,20,26] or not at all [22].

During ECT and in the immediate postoperative period, no serious adverse events were reported for the majority of treated patients, with a few exceptions [19,31]. One patient was reported to die due to myocardial infarction on the third post-operative day [24]. As described in some reports [15,18,20,23,26] and summarized by Plaschke et al. [34], ECT application was followed successively by the swelling, necrotic, and healing phases. Swelling gradually increases one to two days after the procedure and persists for a week or two; it may require elective tracheostomy to avoid airway obstruction when base of tongue, laryngeal, or hypopharyngeal tumors are treated [26,30,31]. The whitening of the treated mucosa during the first days post-ECT signals the necrotic phase, which results in necrosis development in weeks two to six. The healing phase occurs six to nine weeks after ECT and the time course of tissue changes may vary depending on previous treatments and the healing potential of the treated mucosa [15,27,34].

During the immediate postoperative period, the pain peaks 2 h after ECT, followed by a decline 18–24 h later [30]. Later, and if untreated, the pain accompanies the development of necrosis, culminating three to four weeks after ECT [30]. With adequate pain treatment, no change [31,34] or a significant reduction in pain scores [33,35] was recoded between baseline and weeks four or eight post-ECT. Similarly, no major problem with bleeding was reported from the electrode puncture site; moreover, ECT resulted in significant improvements in bleeding control at four weeks post-treatment [33,35]. None of the studies reported any bleomycin toxicity, such as lung fibrosis, allergic reaction, or hematological toxicity.

Late adverse effects of ECT were rare (19 events in 17 patients): above all, their occurrence depended on the site and extent of the treated lesion (Table 5). The results of the quality-of-life assessment by the European Organization for Research and Treatment of Cancer (EORTC) QLQ-C30 and/or QLQ-HN35 questionnaires demonstrated no change from baseline to week four, except less favorable swallowing and social eating scores [31,34]; however, improvement was reported at week eight in well-being and at four months regarding swallowing [31]. At 12 months, no significant difference was found compared to the baseline scores [29]. Similarly, the EuroQuality-5D questionnaire showed no change at four weeks compared to baseline, but showed an increase in well-being at eight weeks and four months post-ECT [31]. In the area of mental health, the results of the Short Form−36 survey reported by four patients treated with ECT for debulking before the definitive therapy showed a marked post-treatment decline one month after ECT, likely due to increased emotional awareness of the disease status [30].

## 4. Discussion

Since the first report on the clinical use of ECT in 1991 [37], only 164 patients with mucosal HNC were reported in the literature who received this treatment (Table 1). However, an increasing trend in its use can also be observed after 2006 when the ESOPE guidelines for the implementation of the ECT procedure for cutaneous tumors were published [5]. After this time point, 72% of all reported cases were treated; it seems that in the last decade, with a share of 58.5% of treated patients and when the two largest and highest quality studies were also conducted, its popularity slowly increased [31,34]. With few exceptions [18,29], other studies have been smaller, some even retrospective [25,26,27], and patients with mucosal head and neck tumors accounted for only a (small) fraction of all ECT-treated cases [17,19,20,22,24,25,26,27,33,35,36]. The quality of reports in terms of the description of the study population and outcomes, and to a lesser extent the treatment delivery, is not optimal and does not allow the identification of all important details [19,22,27,35]. A similar finding was made by Campana et al., who analyzed the quality of reporting in all studies on ECT published between 2006 and 2015 [38]. These shortcomings hinder the performance of meta-analyses or systematic reviews; therefore, the authors provided a summary checklist for data reporting that should improve the quality of reporting clinical ECT studies in the future [38].

The most notable feature of the ECT of mucosal HNC is the limited visibility and accessibility, which undoubtedly explains the limited use of ECT in these tumors (Figure 2). These restrictions require special skills and a more complex procedure than for skin lesions, with the use of surgical aids such as mouth gags for keeping the mouth open, the trans-nasal insertion of a video fiberscope to display the tumor area on the screen, and electrodes attached to forceps to improve the accessibility of difficult-to-reach lesions [14,34]. Moreover, some lesions are accessible only intraoperatively (e.g., by rhinotomy or after the elevation of a skin flap) [16,17]. Patients are usually treated under general anesthesia in order to avoid unpleasant and alarming sensations caused by muscle contraction induced by voltage pulses [17]. However, this adds to the logistic complexity of the procedure that limits the number of ECT treatments, in general and at the level of the individual patient. In total, 88.2% of the reported patients had only one ECT treatment. In the case of oral cavity tumors, a nasal intubation is indicated to improve room for maneuvering and a combined approach to tumors growing in the anterior floor of the mouth by electroporation through the oral mucosa, submental skin, and/or surgical incision in the neck. When swelling of the airway mucosa is anticipated in the early post-operative period (e.g., after ECT on base-of-tongue, laryngeal, and hypopharyngeal tumors), a tracheotomy should be performed before ECT [26,30,31]. Corticosteroid and antibiotic prophylaxis are indicated in all patients to reduce the risk of mucosal swelling and infection to decrease the pain after ECT procedures. The likelihood of severe bleeding, however, is low due to a vascular block elicited by electroporation [8,39].

All 164 mucosal HNCs were treated with bleomycin, mainly administered intravenously (62.5%), especially in more recent studies, assuring a more homogenous distribution of the drug inside the tumor volume, particularly for less accessible and larger lesions. With electroporation, bleomycin cytotoxicity is potentiated several thousand-fold, allowing the use of lower drug doses with minimal systemic toxicity to achieve a significant antitumor effect [7]. With undiminished efficacy in pre-treated tumors and human papillomavirus-positive tumors, bleomycin seems to be preferred over cisplatin-based ECT in mucosal lesions [40,41]. In contrast, no specific recommendation can be made regarding the type of electrodes to be used, as this depends on the specific clinical situation, namely, the accessibility, size, and shape of the lesion. In reported patients, hexagonal electrodes were most commonly used, followed by linear or finger-type electrodes. The ESOPE guidelines were used in all patients treated after 2006 [5,6].

Curative intent ECT was undertaken for 36 patients reported in four studies [18,25,29,30]. Two of them were larger prospective phase 2 studies with 12 and 19 patients with cT1–2 tumors of the oral cavity (27 cases) or oropharynx (four cases) being included [18,29]. The first was the study of Burian et al., which can be considered as a “proof of principle study.” The response was evaluated four weeks post-ECT by a histopathological examination of resected necrotic tissue at the site of the former tumor. It revealed tumor-free tissue in 10 cases; the remaining cancer cells were found at the resection margin and in the center of the necrotic tumor of two cases. No local recurrence was detected during a mean observation period of 10.6 months (range 5–18) [18]. The results of the second study reported by Landström et al. were blurred by RT that followed ECT application in 12 of 19 patients. In seven patients treated solely with ECT, however, local control was 100% at five years, and there was no change in their quality-of-life scores at 12 months after ECT compared to baseline [29]. These results suggest the potential curative capacity of single-course ECT for early-stage tumors of the oral cavity and oropharynx. ECT was used neoadjuvantly before chemoradiation in four locally advanced cT3–4 tumors [30] and, in the case of cT2 soft palate tumors, surgery was successfully employed after PR to a single ECT application [25].

The majority of patients (128, 78%) were treated with palliative intent. In this group, less than a third of the evaluable tumors (in 32 of 106, 30.2%) responded to ECT with CR, which can be long lasting in sporadic cases [17,20,26,31,34]; PR, SD, and PD were recorded in 41.5%, 22.6%, and 5.7% of patients, respectively. The probability of CR after ECT did not correlate with the size (in cm) of the treated lesions but only with T-stage, which also takes into account the relationship of the tumor to the surrounding structures, i.e., eventual ingrowth into the adjacent anatomical (sub)sites. The anatomical complexity of the region does not seem to allow for a simple size–effect relationship in mucosal HNC. However, when the objective response (CR and PR) was considered, no difference was observed across different T-stages, and the cumulative proportion of objective responses of 71.7% was rather comparable to 65% reported in the largest and the most representative EURECA study [31]. Unfortunately, due to the lack of head-to-head comparisons, the efficacy of palliative intent ECT cannot be evaluated in relation to other therapeutic modalities. Moreover, ECT was usually employed as a last alternative, after several unsuccessful previous attempts with standard therapies, which further precludes fair comparisons with other palliative treatment options. Comparing the results of different ECT studies is also problematic due to the vast heterogeneity of treated tumors. This is evident from the size range and also the fact that safety margins around the tumor were treated in some tumors [19,20,24,26,31,34], while other lesions were too extensive and/or difficult to reach for effective electroporation, so that only debulking was performed [31,34].

The absence of serious adverse effects during ECT and in the early post-ECT period and rather infrequent late toxicities speak in favor of the safety of ECT. However, the healing process can be adversely affected by previous treatments, particularly RT, causing fibrosis and vascular damage in treated tissues [15,27,34]. Due to the possibility of mucosal swelling, infection, and pain during a six- to nine-week healing period, appropriate anti-edema, antibiotic, and analgesic prophylaxis and/or therapy is required. The synchronized delivery of electrical pulses with an electrocardiogram as a preventive measure to reduce the risk of arrhythmias is less important in the ECT of these tumors due to their distance from the heart [42]. According to the results of the quality-of-life questionnaires and pain scales, in general one can expect no deterioration from the baseline at weeks four to eight or later after ECT, as well as an improvement in bleeding control [31,33,34,35].

We are aware that the present analysis may include several hidden biases, resulting from the relatively small number of cases analyzed, the poor quality of some studies with incomplete data reporting, and the possibility that not all cases were reported or were presented in formats not considered in this review (i.e., meeting posters or abstracts and course presentations). Furthermore, summarizing the results of several smaller series reported by different institutions over several decades may blur the picture due to differences in the quality of diagnostics and, consequently, in the assessment of the initial tumor size, as well as response to ECT. In this regard, the message from the DAHANCA 32 study is important. The authors found contrast-enhanced CT scanning combined with RECIST criteria and a tissue biopsy for the assessment of treatment response suboptimal, especially if performed less than eight weeks after ECT. They recommended the inclusion of magnetic resonance imaging and/or positron emission tomography in the evaluation of tumor debulking for better visibility of tumor borders and assessment of the viability of residual tumor mass [34].

Various novelties that were recently introduced in the field may have a positive effect on the popularity of ECT in mucosal HNCs. Technical improvements attempt to overcome anatomical disadvantages of the head and neck area by improving the ability to establish an optimal electric field distribution in the volume of interest. To this end, innovative types of electrodes, including endoscopic electroporation systems, and the concept of individual treatment planning have been developed [43,44,45]. The former improves the availability of difficult-to-reach tumors, while the latter allows the optimization of various electroporation parameters according to a given anatomical situation. In this context, the recent advancements in robotic surgery should be taken into account as a competing technique [46]. The use of computer technology enables the 3D display of the target, the calculation of the optimal electrode configuration in the treated volume and various parameters of voltage pulses, and the visualization of the resulting electric field distribution in the tissue [47,48]. The precise implementation of the treatment plan, i.e., electrode insertion, can be image-guided or supported by navigation (i.e., robotic-assisted) [49]. The so-called variable-geometry ECT, with coupling treatment planning and navigation, has been successfully performed for deep-seated lesions in the neck [50]. Furthermore, studies on bleomycin pharmacokinetics have suggested a longer therapeutic window after bleomycin injection for ECT, and have implied the possibility of reducing the bleomycin dose required for effective ECT in elderly patients. This concept has already been confirmed in clinics [51,52,53]. In the era of innovative immunomodulatory drugs, a new paradigm has emerged to upgrade the local effect of ECT into a systemic one. The massive release of tumor antigens and the secretion of damage-associated molecular patterns (e.g., adenosine triphosphate, heat-shock proteins, and calreticulin) and cytokines (e.g., interferon-gamma, interleukin-2, and tumor necrosis factor-alpha) from electroporated tumor cells lead to the recruitment and activation of dendritic- and antigen-presenting cells, resulting in the induction of immunogenic cell death [54]. The combination of ECT and immunotherapy was found to be a promising treatment strategy in retrospective reviews in melanoma patients and is currently being tested in several prospective clinical trials [55,56].

## 5. Conclusions

Preliminary data suggest that ECT may be an effective local treatment option in selected patients with HNC. However, its use is limited, primarily due to the anatomical complexity of the region; consequently, the difficult accessibility of lesions leads to problems in creating a homogeneous and conformal electric field. The development and research in the fields of electrode design, image-guided and robotic-assisted approaches, bleomycin kinetics, and interactions with the immune system give hope that ECT will play a more decisive role in these cancers in the future. Current evidence only justifies the palliative use of ECT in patients without standard local or systemic treatment options. In curative intent treatment scenarios, prospective clinical studies are needed to evaluate ECT compared with existing standard therapies.

## Figures and Tables

**Figure 1 cancers-13-01254-f001:**
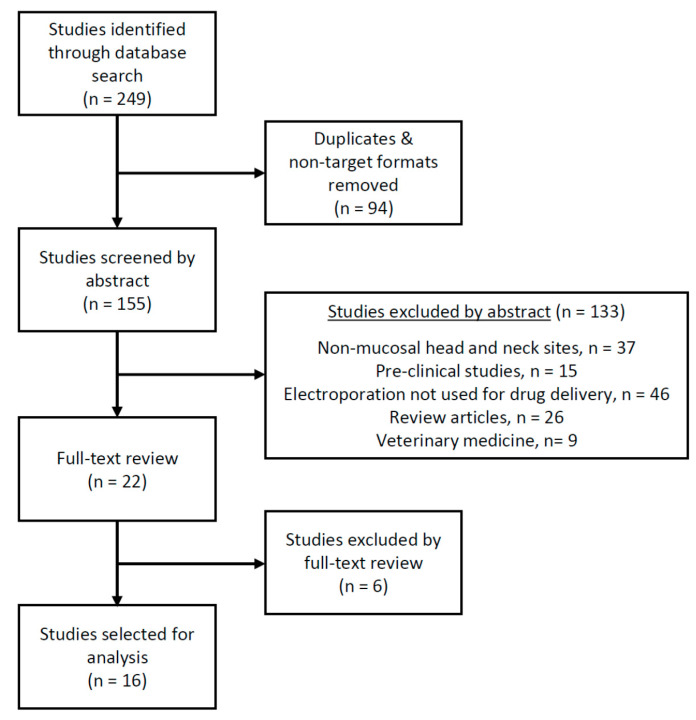
Study selection based on the inclusion and exclusion criteria.

**Figure 2 cancers-13-01254-f002:**
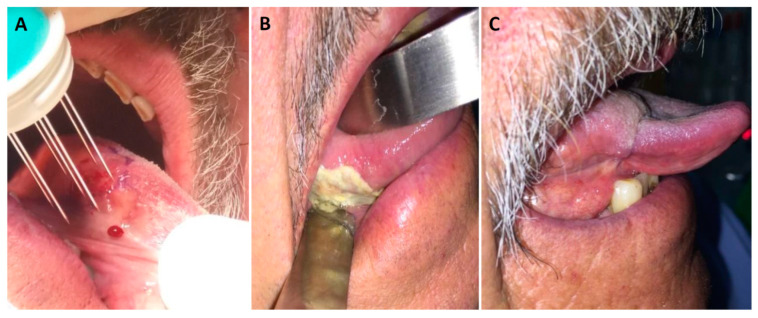
(**A**) A 69-year old patient with cT1N0M0 squamous cell carcinoma on the lateral surface of the mobile tongue. Electric pulses were delivered through hexagonal electrodes 8 min after intravenous injection of bleomycin (15,000 IU/m^2^). (**B**) Necrosis with fibrin plaque formation two weeks after ECT. (**C**) Complete response and scar formation one year after ECT.

**Table 1 cancers-13-01254-t001:** Summary of the characteristics of the included studies.

Study [Ref]	No. of pts.^1^	Study Design	Study Period	Treatment Intent	Response Evaluation		Additional Therapy ^2^
					Time after ECT	Method	
Allegretti et al., 2001 [17]	11/14	Prospective, phase I/II	8/1996–7/1998	Palliative	1 mo	Biopsy	No
Burian et al., 2003 [18]	12/12	Prospective, phase II	n.r.	Curative	4 wks	Surgical resection	Yes, 10/12 (neck dissection)
Bloom et al., 2005 [19]	4/54 ^3^	Prospective, phase II (two multicenter studies)	n.r.	Palliative	n.s.	Clinically	No
Tjink et al., 2006 [20]	2/7	Prospective, case series	n.r.	Palliative	1 and 2 mos	Clinically w/o biopsy	No
Scarlatos et al., 2011 [22]	4/52	Prospective	11/2007–2010	Palliative	2 mos	Clinically	Yes, 15/52 (radiotherapy/surgery)
Mevio et al., 2012 [24]	1/15	Prospective, phase II	4/2009–1/2011	Palliative	4 wks	Radiologically (RECIST)	No
Gargiulo et al., 2012 [25]	2/25	Retrospective	5/2007–9/2010	Curative/palliative	6 wks	Clinically, biopsy	Yes, 3/25 (surgery)
Seccia et al., 2014 [26]	3/9	Retrospective	5/2010–1/2013	Palliative	8 wks	Clinically	No
Campna et al., 2014 [27]	12/42	Retrospective	5/2006–9/2012	Palliative	n.s.	Clinically (RECIST)	No
Landström et al., 2015 [29]	19/19	Prospective, phase II	5/2005–5/2007	Curative	8 wks	Biopsy	Yes, 12/19 (radiotherapy)
Domanico et al., 2015 [30]	4/4	Prospective, phase II	2/2013–2/2014	Curative	4 wks	Clinically (RECIST)	No
Plaschke et al., 2017 [31]	43/43	Prospective, phase II	11/2011–10/2015	Palliative	8 wks	Radiologically w/o biopsy (RECIST)	No
Pichi et al., 2019 [33]	9/36	Prospective, phase II	4/2012–11/2017	Palliative	1 mo	Radiologically (RECIST)	No
Plaschke et al., 2019 [34]	13/26 ^4^	Prospective, phase II	2/2014–9/2017	Palliative	4 and 8 wks	Radiologically, biopsy ^5^ (RECIST)	No
Longo et al., 2019 [35]	24/93	Prospective	5/2011–4/2017	Palliative	8 wks	Radiologically (RECIST)	No
Pichi et al., 2020 [36]	1/1	Case report	4/2019	Palliative	1 mos	Radiologically (RECIST)	No

ECT—electrochemotherapy, n.r.—not reported, n.s.—not specified, RECIST—Response Evaluation Criteria in Solid Tumors. ^1^ With mucosal tumors/All reported. ^2^ Before response evaluation. ^3^ Number of mucosal tumors is not given; side effects of mucosal ECT were described in 4 patients. ^4^ The first 13 patients were reported in Ref. [31].^5^ Biopsies were performed 1 and 4 weeks post-ECT.

**Table 2 cancers-13-01254-t002:** Characteristics of the patients and tumors.

Characteristic	All	Curative Intent	Palliative Intent
No. of patients	164	36	128
Age (years)			
Individual data (*N* = 65); median	65	58.5	68
Group data (*N* = 55); median	57 [18], 68 [31]	57 [18]	68 [31]
Range	20–95	20–78	46–95
n.s. (*N* = 44)	-	-	-
Gender			
Females	34	11	23
Males	86	25	61
n.s.	44	0	44
Histology			
Squamous cell carcinoma	130	35	95
Adenocarcinoma	3	1	2
Adenoid cystic carcinoma	3	0	3
n.s.	28	0	28
Anatomical site			
Oral cavity	103	30	73
Oropharynx	20	6	14
Oral cavity/Oropharynx	19	0	19
Pharynx, n.s.	6	0	6
Nasopharynx	2	0	2
Larynx	10	0	10
Hypopharynx	1	0	1
Nasal cavity/paranasalsinuses	3	0	3
Tumor stage/size (cm)			
T1	11	9	2
T2	29	23	6
T3	4	3	1
T4	17	1	16
Size, *N* = 30; median (range)	2.9 (1.0–14.5)	-	2.9 (1.0–14.5)
n.s.	73	0	73

n.s.—not specified; N—number of patients.

**Table 3 cancers-13-01254-t003:** Characteristics of the electrochemotherapy procedure.

Characteristic	All	Curative Intent	Palliative Intent
No. of tumors	164	36	128
Type of anesthesia			
Local (± sedation)	10	0	10
General	136	36	100
n.s./n.r.	18	0	18
Chemotherapy agent			
Bleomycin	164	36	128
Route of drug administration			
Intratumorally	52	31	21
Intravenously	100	5	95
Combined	8	0	8
n.s.	4	0	4
Electroporator type			
MedPulser (Genetronics Inc., San Diego, CA) ^1^	116	5	111
Cliniporator (IGEA, Carpi, Italy)	48	31	17
Type of electrodes			
Linear	17	0	17
Finger	10	0	10
Hexagonal	86	32	54
Plate	1	0	1
Combined (finger, hexagonal)	5	0	5
n.s.	45	4	41
Safety margin treated			
Yes	78	31	47
No	30	0	30
n.s./n.r.	56	5	51
No. of ECT applications			
1	105	36	69
2	10	0	10
3	4	0	4
n.s.	45	0	45
ESOPE protocol			
Yes	116	5	111
No	48	31	17

n.s.—not specified; n.r.—not reported; ECT—Electrochemotherapy; ESOPE—European Standard Operating. Procedures in Electrochemotherapy.^1^ In the study by Allegreti and Panje (ref. [17]), in addition to the MedPulser (Genetronics Inc., San Diego, CA, USA) two other types of electroporators were used (Electrosquare Porator and 820 RN).

**Table 4 cancers-13-01254-t004:** Palliative electrochemotherapy: tumor response.

Parameter	All ^1^	CR	PR	SD	PD
Responses	106	32	44	24	6
No. of cycles	54				
1	43	15	17	8	3
2	7	2	4	1	0
3	4	1	3	0	0
Histology	65				
Squamous cell carcinoma	63	21	25	14	3
Adenocarcinoma	1	0	1	0	0
Adenoid cystic carcinoma	1	0	1	0	0
Tumor site	57				
Oral cavity	39	13	17	3	6
Oropharynx	10	3	5	2	0
Hypopharynx	2	0	1	1	0
Nasopharynx	2	1	1	0	0
Larynx	1	1	0	0	0
Nasal cavity/paranasal sinuses	3	2	1	0	0
Stage T ^2^	47				
T1–2	23	12	7	3	1
T3–4	24	3	15	5	1
Tumor size	51				
≤3 cm	22	6	11	4	1
>3 cm	29	5	11	11	2

CR—complete response; PR—partial response; SD—stable disease; PD—progressive disease. ^1^ Total number of cases with particular characteristic and tumor response to ECT reported. ^2^ Complete response rates in T1–2 vs. T3–4 tumors: 52.2% vs. 12.5%, *p* = 0.005.

**Table 5 cancers-13-01254-t005:** Long-term serious adverse events after electrochemotherapy.

Study [Ref]	Serious Adverse Event	Tumor Site	Stage T	Time after ECT
Allegretti et al., 2001 [17]	Septum perforation	Nasal septum	T2	n.r.
	Osteomyelitis	Nasopharynx	T4	n.r.
	Dysphagia and bleeding	Base of tongue	T2	n.r.
	Dysphagia	Hard palate, alveolus, maxillary sinus	T4	n.r.
	Dysphagia	Retromolar trigon	T4	n.r.
	Nasocutaneous fistula	Ethmoid sinus	T4	n.r.
	Fistula	Oropharynx	T4	n.r.
Bloom et al., 2005 [19]	Bleeding	Parapharyngeal space	n.r.	3.5 mos
	Cellulitis (of the jaw)	Floor of mouth	n.r.	2 mos
	Cellulitis (of the neck)	Tonsil, base of tongue	n.r.	1 mos
Seccia et al., 2014 [26]	Sepsis	Floor of mouth, mandibular bone, skin	T4	1.5 mos
Landström et al., 2015 [28,29]	Osteoradionecrosis	Floor of mouth	T2	2.5 mos
	Osteoradionecrosis	Floor of mouth	T2	8 mos
	Fistula	Bucca	T1	8 mos
	Bleeding and aspiration	Base of tongue	T2	1.5 mos
Plaschke et al., 2017 [31]	Mucosal edema (tracheostomy dependence)	Hypopharynx	n.r.	n.r.
Plaschke et al., 2019 [34]	Bleeding	Tonsil	n.r.	8 wks

n.r.—not reported.

## Data Availability

The datasets analyzed during the current study are available from the corresponding author on reasonable request.

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
