# Peer review of "Electrochemotherapy in Mucosal Cancer of the Head and Neck: A Systematic Review"

_cancers, 2021, doi:10.3390/cancers13061254_

Round 1

Reviewer 1 Report

a very good systematic enquiry, with good review of a little used technique in head and neck for the reasons described in the paper

Author Response

Reviewer #1

Very good systematic enquiry, with good review of a little used technique in head and neck for the reasons described in the paper.

The authors thank for the positive evaluation of our work.

Reviewer 2 Report

In the review manuscript, the authors mounted the electrochemotherapy against head and neck cancer. The contents are well organized and the authors reveal the deep vision of electrochemotherapy. The quality of the manuscript is high enough to publish without revision. Thus, it needs to be accepted without any further change.

Author Response

Reviewer #2

In the review manuscript, the authors mounted the electrochemotherapy against head and neck cancer. The contents are well organized and the authors reveal the deep vision of electrochemotherapy. The quality of the manuscript is high enough to publish without revision. Thus, it needs to be accepted without any further change.

The authors thank for a very favorable assessment of our work.

Reviewer 3 Report

The paper reviews the available littereture in head and neack tumors treated by means of electrochemotherapy.

Some suggestions:

Some sentences have to be clarified since they are difficult to read.

Line 73 Please, check style and references

Line 149 Please, clarify the sentence ‘initial database search identified 249 records’

In all studies the electroporator was the same? At line 220 ‘hexagonal electrodes´ is reported. Then all the studies refer to the same device for ECT? Please, clarify the electroporation device in the analyzed studies

Lone 243 Please check ‘responders’

Please, be careful to define all the used acronyms

The authors analyzed all the available literature in the field? For instance also the following papers were available in PUBMEd:

The role of electrochemotherapy in the treatment of metastatic head and neck cancer.

De Virgilio A, Fusconi M, Greco A, de Vincentiis M. Tumori. 2013 Sep-Oct;99(5):634. doi: 10.1700/1377.15314.

Electrochemotherapy as a new therapeutic strategy in advanced Merkel cell carcinoma of head and neck region.

Scelsi D, Mevio N, Bertino G, Occhini A, Brazzelli V, Morbini P, Benazzo M. Radiol Oncol. 2013 Oct 8;47(4):366-9. doi: 10.2478/raon-2013-0059. eCollection 2013. PMID: 24294181 Free PMC article.

The value of electrochemotherapy in the treatment of peristomal tumors.

Campana LG, Bertino G, Rossi CR, Occhini A, Rossi M, Valpione S, Benazzo M. Eur J Surg Oncol. 2014 Mar;40(3):260-2. doi: 10.1016/j.ejso.2013.11.013. Epub 2013 Nov 28. PMID: 24332582

Predicting patients at risk for pain associated with electrochemotherapy.

Quaglino P, Matthiessen LW, Curatolo P, Muir T, Bertino G, Kunte C, Odili J, Rotunno R, Humphreys AC, Letulé V, Marenco F, Cuthbert C, Albret R, Benazzo M, De Terlizzi F, Gehl J. Acta Oncol. 2015 Mar;54(3):298-306. doi: 10.3109/0284186X.2014.992546. Epub 2015 Jan 16.

Line 336 ‘electric pulses’ => ‘voltage pulses’. Also line 442

Line 379 please, clarify the sentence ‘In seven patients treated solely with ECT, however, local control was 100%...’

After this rview the authors wha can conclude? ECT could be an option in therapy? please, support by numbers. Moreover,  stress in the discussion or in conclusion the possibility to introduce the ECT in clinical pratice to improve the therapy success of some type of cancer. What tumors could be preferible treated?

Round 2

Reviewer 3 Report

The paper was improved and it is a complete review of the topic.

Only a suggestion

In line 220, we have completed the description of the electrode type: “hexagonal array electrodes”.

hexagonal electrode is correct. hexagonal array is not correct since array identify needle arranged in two parllel lines